# Anomalous quantum Griffiths singularity in ultrathin crystalline lead films

Yi Liu [1,2,6], Ziqiao Wang[1,2,6], Pujia Shan[1,2], Yue Tang[1,2], Chaofei Liu[1,2], Cheng Chen[1,2], Ying Xing[1,2], Qingyan Wang[1,2], Haiwen Liu[3], Xi Lin[1,2], X.C. Xie[1,2] & Jian Wang [1,2,4,5]

Superconductor-insulator/metal transition (SIT/SMT) represents a prototype of quantum phase transition, where quantum fluctuation plays a dominant role and dramatically changes the physical properties of low-dimensional superconducting systems. Recent observation of quantum Griffiths singularity (QGS) offers an essential perspective to understand the subtleties of quantum phase transition in two-dimensional superconductors. Here we study the magnetic field induced SMT in ultrathin crystalline Pb films down to ultralow temperatures. The divergent critical exponent is observed when approaching zero temperature quantum critical point, indicating QGS. Distinctively, the anomalous phase boundary of SMT that the onset critical field decreases with decreasing temperatures in low temperature regime distinguishes our observation from previous reports of QGS in various two-dimensional superconductors. We demonstrate that the anomalous phase boundary originates from the superconducting fluctuations in ultrathin Pb films with pronounced spin-orbit interaction. Our findings reveal a novel aspect of QGS of SMT in two-dimensional superconductors with anomalous phase boundary.

[1] International Center for Quantum Materials, School of Physics, Peking University, 100871 Beijing, China. [2] Collaborative Innovation Center of Quantum Matter, 100871 Beijing, China. [3] Center for Advanced Quantum Studies, Department of Physics, Beijing Normal University, 100875 Beijing, China. [4] CAS Center for Excellence in Topological Quantum Computation, University of Chinese Academy of Sciences, 100190 Beijing, China. [5] Beijing Academy of Quantum Information Sciences, 100193 Beijing, China. [6] These authors contributed equally: Yi Liu, Ziqiao Wang. Correspondence and requests for materials should be addressed to J.W. (email: jianwangphysics@pku.edu.cn) or to H.L. (email: haiwen.liu@bnu.edu.cn)

As a paradigm of quantum phase transition[1,2], superconductor–insulator/metal transition (SIT/SMT) has been widely investigated in two-dimensional (2D) superconductors over the past 30 years[3–11]. When the external magnetic field slightly surpasses the critical value $B_c$ of the quantum critical point, superconductivity will be destroyed and the system enters an insulating state or a weakly localized metal state. In general, a conventional Bardeen–Cooper–Schrieffer-type superconductor has a clear phase boundary with a monotonically negative correlation between the upper critical field and temperature. However, in high temperature and organic superconductors, as well as disordered low-dimensional superconducting systems, the relatively low superfluid density leads to predominant fluctuation effect and as a consequence the phase boundary between superconducting state and normal state becomes indistinct[12–14]. Previous experimental work has shown that the superfluid density of ultrathin Pb films decreases dramatically from the bulk value[15], indicating a significantly larger role of fluctuation in Pb films compared to the bulk counterpart. Moreover, superconducting fluctuation leads to the formation of Cooper pairs in the normal state, which largely influences the characteristic of the phase boundary[16,17]. For instance, the superconducting fluctuation can give rise to a reentrant behavior[18,19], where the sheet resistance of the system first drops with decreasing temperatures and then rises at lower temperatures. Recent in-depth theoretical investigations reveal the full solution of superconducting fluctuation effect in 2D system without spin–orbit interaction (SOI) and figure out the major contribution of fluctuation near the quantum critical point[20,21]. The superconducting fluctuation as well as the special reentrant behavior would result in a complicated and anomalous phase boundary, which has not been fully understood and requires further experimental investigations and theoretical analysis.

Recently, crystalline 2D superconducting systems offer a promising platform to explore unprecedented quantum phenomena[22], such as the observation of Zeeman-protected Ising superconductivity in 2D superconductors with large SOI[23–27] and the emergence of quantum metal state in highly crystalline 2D systems[28–31]. Among them, one striking phenomenon is the quantum Griffiths singularity (QGS) of SMT characterized as a divergent critical exponent $zv$ at zero temperature quantum critical point due to the formation of large rare regions[26,31–36]. QGS has been reported in a wide range of crystalline 2D superconductors, such as 3-monolayer (ML) Ga film[34], 1-ML NbSe$_2$ film[26], LaAlO$_3$/SrTiO$_3$(110) interface[36], gated MoS$_2$, and ZrNCl[31]. These observations of QGS indicate that dissipation and quenched disorder have dramatic effect on the SMT. In the ultralow temperature regime, the quenched disorder leads to large local superconducting islands (rare regions), and the size of these islands increase exponentially when approaching zero temperature[33]. The slow dynamics (such as the relaxation originating from the lowest energy level) of these large superconducting islands give rise to a divergent critical exponent $zv$ in the SMT of low-dimensional superconductors[32,33]. However, the fate of QGS in 2D systems with large SOI and strong superconducting fluctuations remains an open question. Thus researches on QGS of SMT under the influence of strong superconducting fluctuation in a crystalline 2D superconductor with large SOI are highly desired.

In this paper, we report a novel type of QGS in ultrathin crystalline Pb film, which exhibits an anomalous phase boundary of SMT in low temperature regime. The macro-size atomically flat Pb films were epitaxially grown on striped incommensurate (SIC) phase on Si(111) substrate in an ultrahigh-vacuum molecular beam epitaxy chamber. By systematic transport measurement at ultralow temperatures, the ultrathin Pb film undergoes a magnetic field-induced SMT. Scaling analysis shows that the critical exponent of SMT diverges when approaching quantum critical point, as an indication of QGS. However, the onset critical magnetic fields of Pb film determined by the crossing points of magnetoresistance isotherms decrease significantly with decreasing temperature in low temperature regime, which shows pronounced differences from normal QGS. Further theoretical analysis reveals that the anomalous phase boundary of SMT can be quantitatively explained by the superconducting fluctuation in 2D Pb films with large SOI. This anomalous phase boundary leads to the reentrant behavior of $R_s(T)$ curves, and scaling analysis of the SMT within the reentrant region reveals a new type of quantum critical point with anomalous QGS.

## Results

**Anomalous SMT in 4-ML Pb film**. Figure 1 presents the superconducting properties of 4-ML Pb film down to 0.5 K measured in a commercial Physical Property Measurement System (Quantum Design, PPMS-16) with the Helium-3 option. The schematic of the standard four-electrode transport measurement is shown in the inset of Fig. 1a. The superconducting transition occurs at $T_c^{onset} = 9.21$ K indicated by the deviation from linear extrapolation of the normal state resistance $R_n$, which is even higher than the $T_c$ of bulk Pb ($T_c = 7.2$ K). With decreasing temperature, the sheet resistance $R_s$ drops to zero within the measurement resolution at $T_c^{zero} = 5.73$ K (Fig. 1a). Figure 1b reveals the $R_s(T)$ curves measured at different perpendicular magnetic fields. Distinct from the usual SMT with monotonic phase boundary separating the regions of d$R$/d$T < 0$ and d$R$/d$T > 0$, $R_s(T)$ curves in 4-ML Pb films exhibit a pronounced reentrant behavior at lower temperatures (Fig. 1b). Specifically, when applying magnetic field slightly >3.3 T, the sheet resistance firstly decreases with decreasing temperature, reaches the minimum at $T_{min}$ and then rises at lower temperature. This reentrant behavior, including the fact that $T_{min}$ increases with increasing magnetic field, can be quantitatively reproduced after taking the superconducting fluctuation and the influence of large SOI into account (see Supplementary Fig. 1 and Supplementary Note 1 for more details)[16,17]. This pronounced SOI can also give rise to the Zeeman-protected superconductivity with very large in-plane critical field $B_c$ far beyond the Pauli limit[27].

To systematically investigate the anomalous SMT behavior, we measured the temperature-dependent magnetoresistance in details. Interestingly, the $R_s(B)$ curves cross each other in a relatively large transition region instead of a critical point (Fig. 2a and Supplementary Fig. 2). The crossing points of $R_s(B)$ curves at neighboring temperatures are summarized in Fig. 2b as black circles, which show a large transition region from 3.29 to 4.15 T. We also plot the field-dependent onset critical temperature $T_c^{onset}(B)$ in the same figure as blue circles to show the phase boundary at higher temperatures. Here, $T_c^{onset}(B)$ is defined as the temperature where the $R_s(T)$ curve begins to deviate from the linear extrapolation of the normal state resistance. The measured SMT phase boundary in 4-ML Pb film is anomalous since it bends down at low temperatures <2.44 K. The special phase boundary is quantitatively consistent with the theoretical expectation (orange circles) from the plateaus (d$R$/d$T = 0$) of theoretical $R_s(T)$ curves from 3.6 to 4.1 T in Supplementary Fig. 1, confirming that the anomalous SMT behavior can be ascribed to the superconducting fluctuation in 4-ML Pb film. When approaching the quantum critical point, the disorder induced higher-order corrections beyond the fluctuation effect becomes pronounced. Subsequently, the experimental $R_s(T)$ curve at 3.5 T begins to deviate from the theoretical formula of fluctuation effect (as shown in Supplementary Fig. 1c).

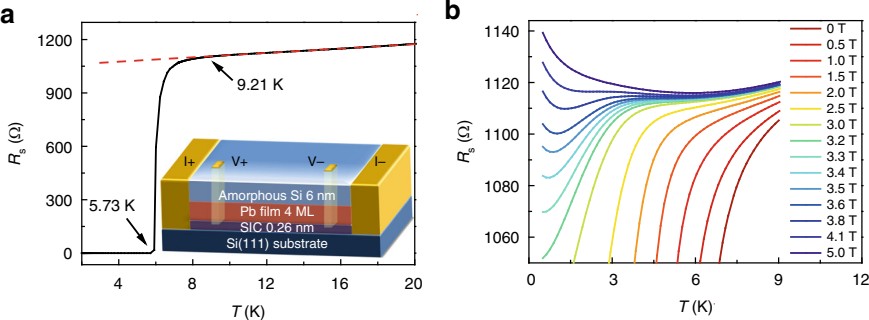

**Fig. 1** Transport properties of 4-monolayer Pb film. **a** Temperature dependence of sheet resistance $R_s$ at zero magnetic field, showing $T_c^{onset} = 9.21$ K and $T_c^{zero} = 5.73$ K. The inset is a schematic for standard four-electrode transport measurements. **b** $R_s(T)$ curves measured under various perpendicular magnetic fields up to 5.0 T, revealing clear superconductor–metal transition and reentrant behavior

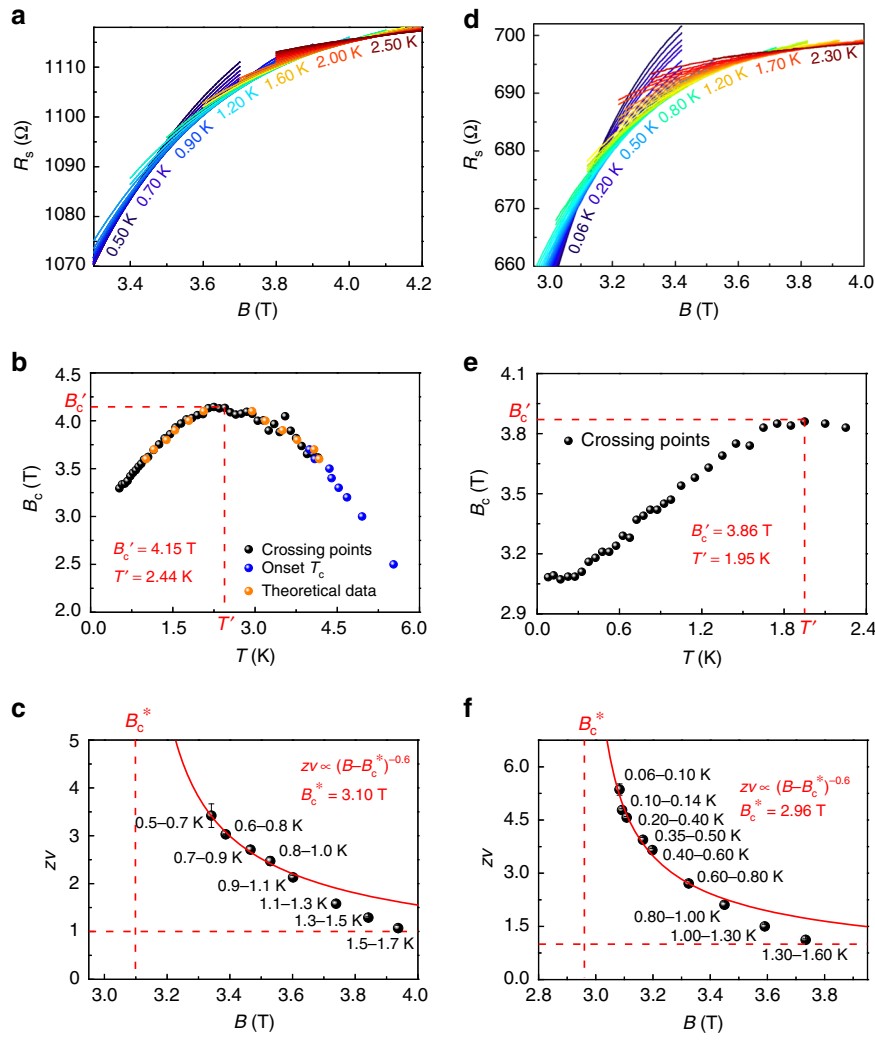

**Fig. 2** Anomalous quantum Griffiths singularity (QGS) of superconductor–metal transition (SMT) in 4-momolayer (ML) Pb films. **a** $R_s(B)$ curves measured in PPMS from 0.5 to 2.5 K, revealing a broad crossing region. **b** The crossing points from neighboring $R_s(B)$ curves (black circles) and onset $T_c$ from $R_s(T)$ curves (blue circles) determine the phase boundary of SMT, which is quantitatively consistent with the superconducting fluctuation theory (orange circles, see Supplementary Fig. 1 and Supplementary Note 1 for more details). **c** Scaling analysis indicates anomalous QGS behavior in 4-ML Pb film. The critical exponent $z\nu$ diverges as $z\nu \propto |B - B_c^*|^{-0.6}$ when temperature approaches zero Kelvin and magnetic field approaches $B_c^*$. **d-f** Similar to **a–c**, but the data were measured down to 60 mK in a dilution refrigerator, exhibiting consistent anomalous QGS behavior

**Anomalous QGS revealed by scaling analysis.** Following the previous work[1,2], we use scaling analysis to determine the critical behavior of the anomalous SMT observed in 4-ML Pb film. $R_s(B)$ curves at neighboring temperatures are classified into one group so that the critical transition region can be approximately regarded as a single critical point $(B_c, R_c)$ (see Fig. 2c, Supplementary Figs. 3 and 4 for details). Thus we can apply the standard finite size scaling in each approximate crossing point to get the effective critical parameters. Generally, the scaling dependence of sheet resistance on temperature and magnetic field takes the form[3,34]: $R_s(B, T) = R_c \cdot F(|B - B_c| T^{-1/z\nu})$, where $F$ is an arbitrary function with $F(0) = 1$, and $z$ and $\nu$ are the dynamical critical exponent and correlation length exponent, respectively. We plot the scaling curves of $R_s(B)/R_c$ at various temperatures against the scaling variable $|B - B_c|t$, where $t = (T/T_0)^{-1/z\nu}$ and $T_0$ is the lowest temperature of the group. Here the parameter $t$ at each temperature $T$ is determined by performing a rescale of scaling variable $|B - B_c|t$ to make the isotherms $R_s(B)/R_c$ at $T$ match the curve at lowest temperature $T_0$ (Supplementary Figs. 3 and 4). The effective critical exponent $z\nu$ is acquired by the linear fitting between $\ln T$ and $\ln t$. For instance, scaling analysis on $R_s(B)$ curves from 500 to 700 mK yields $z\nu = 3.42 \pm 0.25$ (Supplementary Fig. 3b). As shown in Fig. 2c, $z\nu$ grows rapidly and seems to diverge with the temperature decreasing toward zero and the magnetic field tending to a certain critical value $B_c^*$. The divergent behavior of $z\nu$ can be well described by an activated scaling law $z\nu \propto |B - B_c^*|^{-0.6}$[37,38], indicating the existence of QGS with the infinite-randomness quantum critical point (IRQCP) at $B_c^*$ (Fig. 2c). It is noteworthy to mention that, as a result of the anomalous phase boundary, the critical exponent $z\nu$ of anomalous QGS diverges with decreasing field, which is opposite to normal QGS. This uniqueness distinguishes the critical behavior in ultrathin Pb film from previous observations of normal QGS in 2D superconducting systems, including Ga trilayers[34], NbSe$_2$ ML[26], LaAlO$_3$/SrTiO$_3$(110) interface[36], gated MoS$_2$, and ZrNCl[31].

**Ultralow temperature transport measurement.** To confirm the anomalous QGS in ultralow temperature regime, we measured 4-ML sample down to 60 mK in a dilution refrigerator MNK 126–450 system (Leiden Cryogenics BV). Figure 2d shows the $R_s(B)$ curves from 60 mK to 2.30 K and their crossing points are summarized in Fig. 2e. The temperature dependence of critical field $B_c$ in low temperature region is consistent with the data from PPMS measurement. As shown in Fig. 2f, the divergence of critical exponent $z\nu$ can be well fitted using the activated scaling law $z\nu \propto |B - B_c^*|^{-0.6}$ down to 60 mK, providing solid evidence of anomalous QGS in ultrathin Pb film.

**Anomalous QGS in 3.5-ML Pb film.** Figure 3 displays the transport properties of 3.5-ML Pb film under perpendicular magnetic field, which are generally similar to the phenomena observed in 4-ML Pb films. Specifically, $R_s(T)$ curves at various fields have a reentrant behavior at low temperatures (Fig. 3a), and $R_s(B)$ curves at different temperatures reveal a large transition region (Fig. 3b). The SMT phase boundary, which is determined by the crossing points of $R_s(B)$ curves, exhibits an anomalous decrease of critical field $B_c$ with decreasing temperature in low temperature regime (Fig. 3c). Moreover, the critical exponent $z\nu$ also shows a divergent behavior $z\nu \propto |B - B_c^*|^{-0.6}$ when approaching the quantum critical point with $B_c^* = 2.85$ T (Fig. 3d). In the combination of these observations, we confirm the anomalous QGS in 3.5-ML Pb film. As shown in Supplementary Fig. 6, scanning tunneling microscopic (STM) image indicates that the surface disorder of 3.5-ML Pb film is much more pronounced than that of 4-ML Pb film. However, the surface disorder in nanometer scale has little effect on the activated scaling law around the anomalous QGS (Figs. 2 and 3).

**Discussion**

In the ultrathin Pb films, the pronounced superconducting fluctuation and strong SOI effect largely changes the shape of the

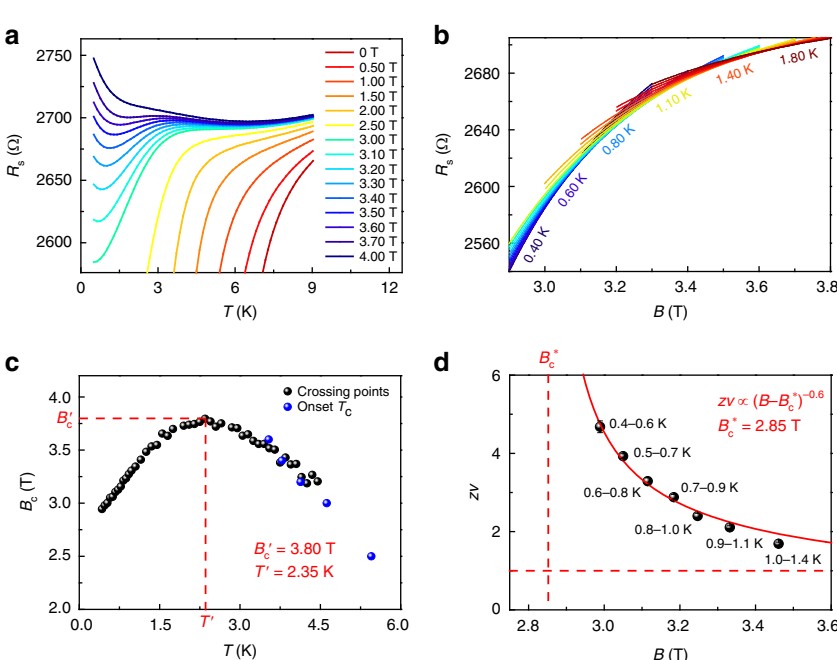

**Fig. 3** Anomalous quantum Griffiths singularity (QGS) of superconductor–metal transition (SMT) in 3.5-monolayer (ML) Pb film. **a** $R_s(T)$ curves at various magnetic fields show SMT and reentrant behavior. **b** $R_s(B)$ curves measured in the temperature region from 0.4 to 1.8 K. **c** The crossing points from neighboring $R_s(B)$ curves and the onset $T_c$ from $R_s(T)$ curves determine the phase boundary of SMT. The phase boundary abnormally bends down at lower temperatures. **d** Scaling analysis indicates anomalous QGS behavior in 3.5-ML Pb film. The critical exponent diverges as $z\nu \propto |B - B_c^*|^{-0.6}$ with temperature approaching zero and magnetic field tending to $B_c^*$

phase boundary. Larkin and his collaborators propose that superconducting fluctuation enhances the conductivity due to the contribution of fluctuating Cooper pairs[16,17]. On the other hand, the formation of Cooper pair reduces the density of states of quasiparticles and decreases the conductivity[12]. Recent in-depth theoretical investigations provide full solution of superconducting fluctuation in 2D system without SOI and also illustrate the real space coherence cluster of fluctuating Cooper pairs[20,21], which may relate to the precursor formation of rare region. Moreover, comparing to previous systems with normal phase boundary[26,31,34,36], the pronounced fluctuation effect in the ultrathin Pb films could be attributed to the relatively low mobility and strong SOI (see Supplementary Note 1 for more details). The SOI leads to the changes in the Cooperon propagator by introducing spin-triplet channel in addition to the spin-singlet channel, which results in different coefficients from those in the previous theory[17] (see Supplementary Note 1 for quantitative fitting of the experimental data). Thus, in ultrathin Pb films, the superconducting fluctuation with strong SOI gives rise to remarkable influence on the conductivity of the system and results in the buckle shape of phase boundary as shown in Fig. 4. When approaching the quantum critical point along this anomalous phase boundary, rare regions of large superconducting islands emerge and dominate the dynamical property of the system (see Supplementary Note 2 for more details). The scaling analysis demonstrates that the activated scaling law $z\nu \propto |B - B_c^*|^{-0.6}$ still holds along this anomalous phase boundary, supporting the universality of QGS around the IRQCP in SMT. Our work suggests that, under the influence of the superconducting fluctuation, the anomalous QGS may exist in various superconducting systems with the reentrant behavior.

Other mechanisms can also give rise to an anomalous phase boundary. First, a strong SOI will bend down the mean field critical field at low temperatures in Werthamer–Helfand–Hohenberg (WHH) theory[39]. However, the WHH with SOI cannot quantitatively reproduce the anomalous phase boundary in our observations. Second, the competition between antiferromagnetism and superconductivity can also give rise to the reentrant behavior[40]. But this mechanism is incompatible with our system, since there has not been any report of antiferromagnetism in crystalline Pb thin film. Lastly, in previous studies of amorphous superconducting films, the reentrant behavior is attributed to the efficient Josephson coupling at high temperatures and the Coloumb blockade effect at low temperatures[41]. However, this process mainly occurs in amorphous or granular systems with large normal resistance, thus ceases to exist in the crystalline Pb thin films.

In summary, the magnetic field-induced SMT is systematically studied in ultrathin crystalline Pb films by ex situ transport measurements. Surprisingly, the phase boundary of SMT exhibits an anomalous behavior in low temperature regime that the onset critical field decreases with decreasing temperatures, which can be quantitatively explained by the effect of superconducting fluctuation with strong SOI. Furthermore, the critical exponent of SMT diverges when approaching zero temperature quantum critical point, indicating the existence of anomalous QGS of SMT in 2D superconducting system showing reentrant superconducting behaviors.

## Methods
**Sample growth**. The ultrathin crystalline Pb (111) films were grown in an Omicron ultrahigh vacuum molecular beam epitaxy chamber at a base pressure $<1 \times 10^{-10}$ mbar. The Si(111)–7 × 7 reconstruction phase was prepared by flashing Si(111) substrate at $T \sim 1400$ K for 5–10 times. Before film growth, the SIC Pb phase was acquired by depositing 1.5-ML Pb from a Knudsen cell at room temperature and then annealing at $T \sim 573$ K for 30 s. The ultrathin Pb (111) films were then grown by depositing Pb atoms on SIC phase at 150 K with a growth rate of

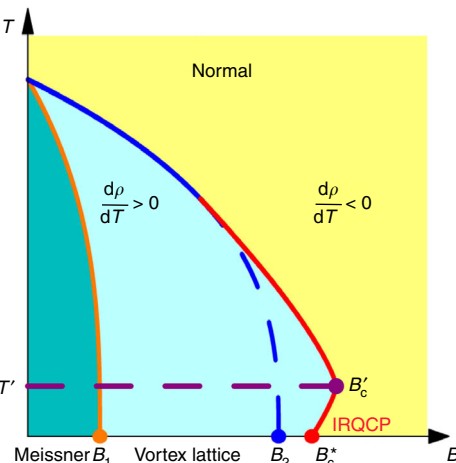

**Fig. 4** Schematic of the $B–T$ phase diagram of two-dimensional superconductor–metal transition (SMT) with pronounced fluctuation effect. $B_2$ is the value of upper critical field with mean field theory. Under the influence of superconducting fluctuations with strong spin–orbit interaction, the mean field phase boundary (blue dashed line) buckles outward to the solid red line, which gives rise to the reentrant phenomena shown in Figs. 1b and 3a. The anomalous phase boundary exists in the regime $[B_c^*, B_c']$. When approaching the real infinite-randomness quantum critical point $B_c^*$ along this anomalous phase boundary, the system exhibits anomalous quantum Griffiths singularity behavior

~0.2-ML/min and subsequent annealing at room temperature for several minutes. Film growth was monitored by reflection high-energy electron diffraction and characterized by STM.

**Transport measurement**. For ex situ transport measurements, Pb films were protected by depositing 6-nm-thick amorphous Si capping layer before exposure to the atmosphere. The resistance and magnetoresistance were measured using the standard four-probe method in a commercial Physical Property Measurement System (Quantum Design, PPMS-16) with the Helium-3 option for temperatures down to 0.5 K under perpendicular magnetic field. The ultralow temperature experiment was carried out in a Dilution Refrigerator MNK 126–450 system (Leiden Cryogenics BV) down to 60 mK. Standard low-frequency lock-in technique was used during the measurements with a current excitation of 50 nA at 23 Hz for ultralow temperature measurements. It is noteworthy to mention that the ultralow temperature measurements (Fig. 2d–f) were carried out right after the growth of sample, so the exposure time to atmosphere is shorter and the sheet resistance is smaller compared to the data from PPMS (Figs. 1, Fig. 2a–c).

## Data availability
The data that support the findings of this study are available from the corresponding authors on reasonable request.

## Code availability
The code in this work is available from the corresponding authors on reasonable request.

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

## Acknowledgements

We thank Hailong Fu and Pengjie Wang for help in ultralow temperature transport measurement. This work was financially supported by the National Key Research and Development Program of China (Grant Nos. 2018YFA0305604, 2017YFA0303300, 2015CB921102, 2017YFA0304600), the National Natural Science Foundation of China (Grant Nos.11888101, 11774008 and 11674028), the Strategic Priority Research Program of Chinese Academy of Sciences (Grant No. XDB28000000), Beijing Natural Science Foundation (Z180010) and China Postdoctoral Science Foundation (Grant No. 2019M650290). H.L. also acknowledges support from the Fundamental Research Funds for the Central Universities.

## Author contributions

J.W. conceived and instructed the research. Z.W., C.L. and Q.W. grew the samples. Y.L. and Z.W. performed the transport measurements in PPMS and analyzed the experimental results. P.S., Y.L. and X.L. performed the transport measurements in dilution refrigerator. Y.L., H.L. and X.C.X. proposed and developed the theoretical model. C.C., Y.T. and Y.X. helped in the sample growth and transport measurements. Y.L., H.L. and Z.W. wrote the manuscript under the supervision of J.W.

## Additional information

**Competing interests:** The authors declare no competing interests.

