## [Peer Review File · Nature Communications]

Reviewers' comments:

Reviewer #1 (Remarks to the Author):

The manuscript NCOMMS-18-31904 by Liu et al reports the observation of anomalous quantum Griffiths singularity (QGS) in ultrathin crystalline lead films and attributes its origin to the superconducting fluctuation effect. The result is very interesting, which expands the boundary of understanding the QGS in a 2D crystalline thin film with strong superconducting fluctuation. Therefore, I recommend it for publication in Nature Communications after clarifying my comments/questions below.

- Can the authors clarify the physical meaning of equations (2) and (3) and the associated parameter, $\psi(r)$, in the supplementary Note 1?

- In Ref. [29], quantum Griffiths states are believed to consist of superconducting puddles and quantum fluctuation also plays an important role in the superconductor-insulator phase transition, however, reentrant phenomenon and the anomalous QGS reported here were not observed in their material systems. It would be helpful to discuss the possible reason and give a schematic image of quantum Griffiths state that is different from Ref. [29].

- For the Supplementary Figure 5 (a), how different in the film thickness for the dark yellow and dark brown regions?

Reviewer #2 (Remarks to the Author):

This paper by Y. Liu, J. Wang and their collaborators discuss the magnetoresistance in crystalline Pb ultrathin films, which show reentrant phenomena. The new finding in this work is the observation of the quantum phase transition with the characteristics of the quantum Griffiths singularity, that is the divergence of the critical exponent, even in the case that the critical field decrease with decreasing temperature.

The data is clear and new. The claim for the Griffiths singularity in this particular case seems to be attractive, since the data possibly shows the universality and extensibility of the quantum Griffiths singularity in the crystalline 2D superconductors. However, there are many questions and problems in the analyses and understanding of the data using the superconducting fluctuation as follows.

(1) Although there has been a variety of reports for the observation of the quantum Griffiths singularity in 2D superconductors, the anomalous Griffiths phenomena with reentrant behavior of resistance seems to be rare. Nevertheless, it is not obviously discussed why this anomalous behavior or the reentrant transition can be observed in Pb ultrathin films. The author attribute it to the effect of superconducting thermal fluctuation, which becomes pronounced due to the low mobility of electrons (or quasi-particles). In general, the effect of superconducting fluctuation is large in 2D systems. The previous 2D systems showing the Griffiths singularities, which contain the authors groups' samples (ref. 24, 32, 34), have a similar sheet resistance around 1000 ohms, Therefore, as a origin of the anomalous behavior, the pronounced fluctuation caused by the low mobility seems inadequate.

(2) For theoretical reproduction of the observed $R(T)$ in magnetic field, the authors use the Aslamazov-Larkin-type fluctuation theory. At first, it is difficult to understand how the authors calculate the $R(T)$ curves in Supplementary Fig. 1(b) from eq. (4) in Supplementary Information . If

the eq. (4) shows the correction term by thermal fluctuation, the observed $R(T)$ curves can be only reproduced by adding the inverse resistance in the normal state. However, it is not clear how they estimate the temperature dependent resistance in the normal state.

(3) In the original theory of ref. 17, it is suggested that α and β in eqs. (1) and (4) in Supplementary Information are $-2/3$ and $8/3$, respectively. I cannot find validity for the values of $\alpha = 1/3$ and $\beta = 0$ the authors used in this manuscript. In addition, why the background conductance σ_0 is necessary in the correction terms? (does it mean the normal state conductance?).

(4) The DOS term in eq. (4) seems quite different from the expression known in general (e.g. Phys. Rev. B 84, 104510 (2011)). Is it natural to use the activated term instead of usual DOS term?

(5) Even if the procedure to calculate the $R(T)$ is correct, the data in Supplementary Fig. 1(a) and the calculated result in Supplementary Fig. 1(b) does not seem to match completely. In this situation, how the author determined the fitting parameters in Supplementary Table 1? If they determine these parameters to match the local maxima and local minima, I think that the plots of theoretical data in Fig. 2(b) and Fig. 3(c) have no meaning. In addition, some parameters in Supplementary Table 1 depend on magnetic field. A valid explanation on the variation of them with magnetic field may be necessary.

(6) After clarifying the above questions, there remains a problem. In the case that $R(T)$ can be reproduced by the thermal fluctuation theory, B_c with the minima in $R(T)$ is not the boundary between the superconducting state and weakly localized metal state, but is simply determined by the balance between the two kinds of terms in eq. (4), which decreases and increases with decreasing temperature, respectively. On the other hand, the authors performed finite size scaling analyses based on the quantum fluctuation model by using B_c . The fact that the electrical resistance in the same temperature and magnetic field regions is analyzed by different kinds of approaches, thermal and quantum fluctuation, may cause contradiction.

(7) The reentrant behavior of the resistance has been already reported in the amorphous films (ref. 6) as well as TiN films (ref 39). In addition to the low mobility as the authors indicate, the similarities and the differences between the precious reports and the present data should be discussed. At least, in ref.6, the homogeneity of the sample is stressed, which seems to be a common feature.

(8) In this manuscript, the authors only show the magnetoresistance near the onset. Many readers would like to know the whole picture of resistive transition in magnetic fields.

Reviewer #3 (Remarks to the Author):

The manuscript "Anomalous quantum Griffiths singularity in ultrathin crystalline lead films" contains interesting experimental data. These data are interpreted using a theoretical concept of quantum Griffiths singularity. The problem is that this phenomenon probably does not exist in disordered superconducting films. One can find a discussion of the situation in the recent review <https://arxiv.org/abs/1712.07215>

On the other hand, it is possible that in an intermediate temperature range experimental data can mimic quantum Griffiths singularity. In my opinion this question should be clarified before publishing of the article.

In the following, the referee's original comments are shown by blue italic characters.
Our responses are shown by black characters.

Reviewer #1 (Remarks to the Author):

***Comment:** The manuscript NCOMMS-18-31904 by Liu et al reports the observation of anomalous quantum Griffiths singularity (QGS) in ultrathin crystalline lead films and attributes its origin to the superconducting fluctuation effect. The result is very interesting, which expands the boundary of understanding the QGS in a 2D crystalline thin film with strong superconducting fluctuation. Therefore, I recommend it for publication in Nature Communications after clarifying my comments/questions below.*

Response: We gratefully thank Reviewer #1 for his/her positive comments and recommendation for the publication of our work. Our manuscript has been revised to incorporate the very helpful comments and suggestions of Reviewer #1. In the following, we give our detailed reply to Reviewer #1's comments.

***Comment:** (1) Can the authors clarify the physical meaning of equations (2) and (3) and the associated parameter, $\psi(r)$, in the supplementary Note 1?*

Response: We thank Reviewer #1 for raising this issue. In our work, the function $\psi(r)$ denotes the digamma function, $I_\alpha(b, t)$ and $I_\beta(b, t)$ in equations (2) and (3) in the supplementary Note 1 are mathematical formulas obtained by calculating the conductivity corrections from the fluctuation theory of superconductor (mainly based on the work of Galitski and Larkin, *Phys. Rev. B* **63**, 174506 (2001), and also highly related to *Phys. Rev. B* **84**, 104510 (2011)).

One major difference of our system with the previous theory (*Phys. Rev. B* **63**, 174506 (2001), *Phys. Rev. B* **84**, 104510 (2011)) is that the Pb films have large spin-orbit interaction (SOI), which gives rises to the Zeeman-protected superconductivity in this system (*Phys. Rev. X* **8**, 021002 (2018)). The SOI effect

alters the Cooperon propagator, and subsequently changes the coefficients in the formula of fluctuation theory. Thus, in the revised manuscript we take all these effects into consideration by conductivity correction $\delta\sigma = \frac{e^2}{\pi^2\hbar} [\alpha I_\alpha(b, t) + \beta I_\beta(b, t)]$ including both the Aslamazov-Larkin term and the Maki-Thomason type term (*Phys. Rev. B* **63**, 174506 (2001)). The SOI makes the parameters α and β of the superconducting fluctuation in the ultrathin Pb films different from the previous theoretical results without SOI (*Phys. Rev. B* **63**, 174506 (2001)). Meanwhile, the formation of fluctuating Cooper pair reduces the number of quasiparticles and decreases the conductivity, known as the density of states term. Combining these effects together, the theoretical formula gives quantitative fitting of experimental data as shown in Fig. R1 with fitting parameters shown in Table R1. We want to mention that α and β are relatively stable at different magnetic fields, which accounts for the reliability of theoretical consideration.

Comment: (2) In Ref. [29], quantum Griffiths states are believed to consist of superconducting puddles and quantum fluctuation also plays an important role in the superconductor-insulator phase transition, however, reentrant phenomenon and the anomalous QGS reported here were not observed in their material systems. It would be helpful to discuss the possible reason and give a schematic image of quantum Griffiths state that is different from Ref. [29].

Response: We thank Reviewer #1 for his/her helpful suggestion. The anomalous reentrant behavior in ultrathin Pb films can be understood as a result of the SOI and disorder-enhanced superconducting fluctuation, which becomes pronounced due to two possible reasons. Firstly, the 4-ML Pb film has a relatively low mobility of 0.064 cm²/Vs in the low temperature regime (around 10 K), which is smaller than that of 3-ML Ga film (0.1 cm²/Vs), 1-ML NbSe₂ film (30 cm²/Vs) and LaAlO₃/SrTiO₃(110) interface (112 cm²/Vs) showing quantum Griffiths singularity. Thus, the superconducting fluctuation in this material becomes more pronounced. Secondly, the SOI can quantitatively affect the superconducting fluctuations. The SOI gives rise to

the changes in the Cooperon propagator by introducing spin-triplet channel in addition to the spin-singlet channel, which influences the parameters α and β from previous theory (*Phys. Rev. B* **63**, 174506 (2001)). Thus, the difference of our work with Ref. [29] might originate from the difference in the explicit values of SOI and mobility in these two systems. To sum up, under the influence of disorder and SOI, the superconducting fluctuation in the ultrathin Pb films results in the buckle shape of phase boundary around the quantum critical point. When approaching the quantum critical point along the anomalous phase boundary, rare regions of large superconducting islands emerge and dominate the dynamical property of the system, which gives rise to the anomalous quantum Griffiths singularity.

To quantitatively investigate the influence of superconducting fluctuation in the ultrathin Pb film, we improve the theoretical model of the superconducting fluctuation in the revised manuscript by providing theoretical fitting of non-monotonic $R_s(T)$ curves (sheet resistance versus temperature) under different magnetic fields. As shown in Fig. R1, the experimental $R_s(T)$ curves can be well fitted by the improved formula, providing evidence for the effect of superconducting fluctuation. More details of the fitting procedure are added in the Supplementary Note 1 in the revised manuscript.

Fig. R1 (a) The experimental $R_s(T)$ curves and the theoretical fitting curves of 4-ML Pb film from 3.6 T to 4.1 T. The theoretical curves are quantitatively consistent with the experimental data of 4-ML Pb film. (b) Normal state resistance $R_n(T)$ curves for theoretical fitting at various magnetic fields are estimated by taking account of the

experimental $R_s(T)$ curve at 5 T (much higher than the superconducting critical field).

Comment: (3) For the Supplementary Figure 5 (a), how different in the film thickness for the dark yellow and dark brown regions?

Response: We thank Reviewer #1 for raising this question. As shown in the revised supplementary Fig. 6 and Fig. R2, the dark yellow region is about 1 monolayer (0.28 nm) thicker than the dark brown regions.

Fig. R2 Typical scanning tunneling microscopy images of (a) 3.5-ML and (b) 4-ML Pb film (*Physical Review X* **8**, 021002 (2018)). The thickness of the islands on 3.5-ML Pb film is one monolayer (around 0.28 nm).

Reviewer #2 (Remarks to the Author):

***Comment:** This paper by Y. Liu, J. Wang and their collaborators discuss the magnetoresistance in crystalline Pb ultrathin films, which show reentrant phenomena. The new finding in this work is the observation of the quantum phase transition with the characteristics of the quantum Griffiths singularity, that is the divergence of the critical exponent, even in the case that the critical field decrease with decreasing temperature.*

The data is clear and new. The claim for the Griffiths singularity in this particular case seems to be attractive, since the data possibly shows the universality and extensibility of the quantum Griffiths singularity in the crystalline 2D superconductors. However, there are many questions and problems in the analyses and understanding of the data using the superconducting fluctuation as follows.

Response: We gratefully thank Reviewer #2 for the careful reading of our manuscript and for his/her positive comment on our finding of anomalous quantum Griffiths singularity. Moreover, we are also very grateful for Reviewer #2 to draw our attention to a highly related reference (*Phys. Rev. B* **84**, 104510 (2011)), which not only provides full solution of superconducting fluctuation but also demonstrates a possible illustration for the real space rare region (Fig. 4 in *Phys. Rev. B* **84**, 104510 (2011)) responsible for the quantum Griffiths singularity. We also thank Reviewer #2 for his/her helpful comments and insightful suggestions on our work. We have revised our manuscript to incorporate the comments from all Reviewers.

In light of the intellectual stimulation from Reviewer #2's valuable comments, we have improved our theoretical model of the superconducting fluctuation, and clarified the effect of spin-orbit interaction (SOI) on the superconducting fluctuation, which gives a quantitative fitting of the experimental $R_s(T)$ curves. One phenomenological difference of our system with the previous theory (*Phys. Rev. B* **63**, 174506 (2001), *Phys. Rev. B* **84**, 104510 (2011)) is that Pb film has large SOI, which gives rises to the Zeeman-protected superconductivity in this system (*Phys. Rev. X* **8**,

021002 (2018)). The SOI effect alters the Cooperon propagator, and can subsequently change the coefficients in the formula of fluctuation theory. In the following, we give our detailed reply to all the comments and suggestions of Reviewer #2.

Comment: (1) Although there has been a variety of reports for the observation of the quantum Griffiths singularity in 2D superconductors, the anomalous Griffiths phenomena with reentrant behavior of resistance seems to be rare. Nevertheless, it is not obviously discussed why this anomalous behavior or the reentrant transition can be observed in Pb ultrathin films. The author attribute it to the effect of superconducting thermal fluctuation, which becomes pronounced due to the low mobility of electrons (or quasi-particles). In general, the effect of superconducting fluctuation is large in 2D systems. The previous 2D systems showing the Griffiths singularities, which contain the authors groups' samples (ref. 24, 32, 34), have a similar sheet resistance around 1000 ohms, Therefore, as a origin of the anomalous behavior, the pronounced fluctuation caused by the low mobility seems inadequate.

Response: We thank Reviewer #2 for pointing out this important issue. The anomalous reentrant behavior in ultrathin Pb films can be considered as a result of the superconducting fluctuation, which becomes pronounced due to the low mobility and pronounced SOI in this system. Firstly, the low mobility in ultrathin Pb films enhances the effect of superconducting fluctuation. The 4-ML Pb film has a relatively low mobility of $0.064 \text{ cm}^2/\text{Vs}$ in the low temperature regime (around 10 K), which is smaller than that of 3-ML Ga film ($0.1 \text{ cm}^2/\text{Vs}$), 1-ML NbSe₂ film ($30 \text{ cm}^2/\text{Vs}$) and LaAlO₃/SrTiO₃(110) interface ($112 \text{ cm}^2/\text{Vs}$) showing quantum Griffiths singularity. Secondly, the SOI alters the form of Cooperon propagator, introducing the spin-triplet channel with negative sign in addition to the spin-singlet channel with positive sign (G. Bergmann, *Physics Report* **107**, 1 (1984)), and thus largely change the contribution of the Maki-Thomason (MT) type terms (MT term and diffusion coefficient renormalization (DCR) terms in the notation of *Phys. Rev. B* **84**, 104510 (2011)). Hence, under the influence of SOI, the superconducting fluctuation can give rise to anomalous reentrant behavior in the $R_s(T)$ curves (resistance versus

temperature) of ultrathin Pb films. (The ultrathin Pb films have large spin-orbit interaction (SOI), which gives rises to the Zeeman-protected superconductivity in this system (*Phys. Rev. X* **8**, 021002 (2018)). We give quantitative fitting of the $R_s(T)$ curves in the following Fig. R1. We have added the related discussion in the revised Supplementary Note 2.

Comment: (2) For theoretical reproduction of the observed $R(T)$ in magnetic field, the authors use the Aslamazov-Larkin-type fluctuation theory. At first, it is difficult to understand how the authors calculate the $R(T)$ curves in Supplementary Fig. 1(b) from eq. (4) in Supplementary Information . If the eq. (4) shows the correction term by thermal fluctuation, the observed $R(T)$ curves can be only reproduced by adding the inverse resistance in the normal state. However, it is not clear how they estimate the temperature dependent resistance in the normal state.

Comment: (3) In the original theory of ref. 17, it is suggested that α and β in eqs. (1) and (4) in Supplementary Information are $-2/3$ and $8/3$, respectively. I cannot find validity for the values of $\alpha = 1/3$ and $\beta = 0$ the authors used in this manuscript. In addition, why the background conductance σ_0 is necessary in the correction terms? (does it mean the normal state conductance?).

Response: We thank Reviewer #2 for raising these insightful comments. We are also grateful for Reviewer #2 for drawing our attention to a highly related theoretical work which provides full solution for the two-dimensional superconducting fluctuation (*Phys. Rev. B* **84**, 104510 (2011)). Based on the in-depth investigation in this work (Table 1, Appendix B and D in *Phys. Rev. B* **84**, 104510 (2011)), the main contribution of superconducting fluctuation around $B_{c2}(T=0)$ is the Maki-Thompson (MT)-type terms (MT and DCR terms in the notation of *Phys. Rev. B* **84**, 104510 (2011)), which all relate to the Cooperon propagator. In the ultrathin Pb films, there exists pronounced SOI, which gives rises to the Zeeman protected superconductivity (*Phys. Rev. X* **8**, 021002 (2018)). The SOI also alters the form of Cooperon propagator by introducing spin-triplet channel. Thus, the values of α and β can be

different from results in previous literatures (*Phys. Rev. B* **84**, 104510 (2011); *Phys. Rev. B* **63**, 174506 (2001)).

In the following, we consider the influence of SOI on the Aslamazov-Larkin (AL) term and the MT-type terms (MT and DCR terms in the notation of *Phys. Rev. B* **84**, 104510 (2011)). The superconducting fluctuation effect on the conductivity correction can be written in the form (*Fiz. Tverd. Tela (Leningrad)* **10**, 1104 (1968), *Phys. Rev. B* **63**, 174506 (2001)):

$$\delta\sigma = \frac{e^2}{\pi^2\hbar} [\alpha I_\alpha(b, t) + \beta I_\beta(b, t)], \quad (\text{R1})$$

with

$$I_\alpha(b, t) = \ln \frac{r}{b} - \frac{1}{2r} - \psi(r), \quad (\text{R2})$$

and

$$I_\beta(b, t) = r\psi'(r) - \frac{1}{2r} - 1, \quad (\text{R3})$$

where $r = \frac{b}{3.562t}$, $t = T/T_c \ll 1$, $b = [B - B_{c2}(T)]/B_{c2}(0) \ll 1$, $\psi(r)$ is the digamma function and $B_{c2}(T)$ is given by the Werthamer-Helfand-Hohenberg theory (*Phys. Rev.* **147**, 295 (1966)). Due to the influence of SOI on the Cooperon propagator, we consider that α and β can be different from the previous result without SOI (*Phys. Rev. B* **63**, 174506 (2001)). And the fitting procedure gives the value of α and β .

Furthermore, the formation of fluctuating Cooper pair can decrease the density of states (DOS). In our simulation, we consider that the influence on the DOS can be represented by a thermally activated behavior of quasiparticles. In the high magnetic field regime (much higher than the superconducting critical field), the contribution of the superconducting fluctuation becomes negligible and the temperature dependence of sheet resistance can represent the normal state resistance. Then we estimate the normal state conductance at different fields by $\sigma_n = \sigma_0 + d \cdot \sigma_{5T}$, where σ_{5T} is the experimental sheet conductance of 4-ML Pb film at 5 T, and σ_0 and d are temperature independent parameters. Therefore, the total conductivity of the system can be written as:

$$\sigma = \sigma_n + \frac{e^2}{\pi^2 \hbar} [\alpha I_\alpha(b, t) + \beta I_\beta(b, t)] + C \left[\exp\left(-\frac{\Delta}{k_B T}\right) - 1 \right], \quad (\text{R4})$$

where Δ is the activation energy of thermal excitation (the local superconducting pairing strength). The discussion related to the DOS term is clarified in the following reply to comment (4).

The experimental $R_s(T)$ curves from 3.6 T to 4.1 T can be well fitted by Eq. (R4) (shown in Fig. R1) with the parameters summarized in Table R1. The normal state resistance is plotted in Fig. R1, which exhibits the behavior of weakly localized metallic state. And considering the effect of SOI on the AL term and MT-type terms (MT and DCR terms in the notation of *Phys. Rev. B* **84**, 104510 (2011)), the parameters α and β are around 0.5 and -0.2, respectively, as shown in Table R1. The value of α and β only slightly depend on the magnetic field, which accounts for the reliability of our consideration. For the DOS term, with increasing magnetic field from 3.6 T to 4.1 T, the local pairing strength Δ remains stable and the coefficient C decreases. It is reasonable that the influence of DOS term (this term takes into account the decrease of quasiparticle DOS due to formation of local fluctuating Cooper pairs) becomes smaller at higher magnetic fields since the fluctuating Cooper pairs are also reduced.

Moreover, considering the influence of SOI, the explicit theoretical formula of the full solution of superconducting fluctuation might be derived in the similar procedure with previous literatures (*Phys. Rev. B* **63**, 174506 (2001), *Phys. Rev. B* **84**, 104510 (2011) and *Rev. Mod. Phys.* **90** 015009 (2018)). The analysis in our work may act as a phenomenological warm-up for this in-depth investigation.

In the revised manuscript, we have added the related references (*Phys. Rev. B* **84**, 104510 (2011) and *Rev. Mod. Phys.* **90** 015009 (2018)), related discussion and detailed fitting results to substantially enhance our theoretical analysis.

Fig. R1 (a) The experimental $R_s(T)$ curves and the theoretical fitting curves of 4-ML Pb film from 3.6 T to 4.1 T. The theoretical curves are quantitatively consistent with the experimental data of 4-ML Pb film. (b) Theoretical normal state resistance $R_n(T)$ curves at various magnetic fields.

Table R1 The fitting parameters at different magnetic fields from 3.6 T to 4.1 T

B (T)	α	β	C (Ω^{-1})	Δ/k_B (K)	σ_0 (Ω^{-1})	d
3.6	0.518	-0.233	9.71×10^{-5}	10.2	5.11×10^{-4}	0.519
3.7	0.518	-0.218	9.47×10^{-5}	10.3	5.00×10^{-4}	0.529
3.8	0.525	-0.205	9.33×10^{-5}	10.2	4.51×10^{-4}	0.583
3.9	0.527	-0.194	9.09×10^{-5}	10.2	3.80×10^{-4}	0.660
4.0	0.528	-0.195	8.72×10^{-5}	10.3	2.92×10^{-4}	0.755
4.1	0.515	-0.171	8.27×10^{-5}	10.3	2.57×10^{-4}	0.789

Comment: (4) The DOS term in eq. (4) seems quite different from the expression known in general (e.g. *Phys. Rev. B* **84**, 104510 (2011)). Is it natural to use the activated term instead of usual DOS term?

Response: We thank Reviewer #2 for raising this issue. We are also grateful for Reviewer #2 for drawing our attention to the highly related theoretical works which provides full solution for the two-dimensional superconducting fluctuation (*Phys. Rev. B* **84**, 104510 (2011) and *Rev. Mod. Phys.* **90**, 015009 (2018)). In order to give quantitative fitting of $R_s(T)$ curves in the relatively high temperature regime, we

choose thermally activated term to represent the density of states (DOS) term, and this term takes into account the decrease of quasiparticle DOS due to formation of local fluctuating Cooper pairs above the phase boundary where long range coherence disappears (the parameter C is used to weigh the amount of these quasiparticles). From the fitting parameter Δ shown in Table R1, we can find this DOS correction mainly accounts for the contribution of high temperature ($T > 1.5$ K). Furthermore, even if the activated DOS term remains a temperature-independent constant, we can still have a fairly good fitting to the reentrant behaviors of the $R_s(T)$ curves for temperature below 1.5 K (Fig. R3) with the same parameters in Table R1, indicating that the anomalous phase boundary mainly results from the AL-type and MT-type (MT and DCR terms in the notation of *Phys. Rev. B* **84**, 104510 (2011)) superconducting fluctuations. To sum up, although this simple thermally activated term may seem to be different from the DOS term in the previous in-depth theoretical results of two-dimensional superconducting fluctuation (*Phys. Rev. B* **84**, 104510 (2011)), it does help us have a better fitting in the relatively high temperature regime and thus gives a complete phase boundary in the relatively high temperature regime.

Fig. R3 The experimental $R_s(T)$ curves and the theoretical fitting curves of 4-ML Pb film from 3.6 T to 4.1 T. The fitting curves are calculated using Eq. (4) with the activated DOS term remains a constant.

Comment: (5) Even if the procedure to calculate the $R(T)$ is correct, the data in

Supplementary Fig. 1(a) and the calculated result in Supplementary Fig. 1(b) does not seem to match completely. In this situation, how the author determined the fitting parameters in Supplementary Table 1? If they determine these parameters to match the local maxima and local minima, I think that the plots of theoretical data in Fig. 2(b) and Fig. 3(c) have no meaning. In addition, some parameters in Supplementary Table 1 depend on magnetic field. A valid explanation on the variation of them with magnetic field may be necessary.

Response: We thank Reviewer #2 for pointing out this issue. According to the insightful suggestion of Reviewer #2, we have improved the theoretical model in the revised manuscript and the fitting curves are quantitatively consistent with the experimental $R_s(T)$ data (Fig. R1). Supplementary Table 1 and Table R1 summarize the fitting parameters, which are determined by matching the theoretical and experimental $R_s(T)$ curves. Considering the SOI effect on the superconducting fluctuation, including the AL term and MT-type terms, the parameters α and β are around 0.5 and -0.2, respectively (Table R1). The value of α and β only slightly depend on the magnetic field, which accounts for the reliability of our consideration. For the DOS term (mainly contributing to conductivity correction of relatively high temperature regime), with increasing magnetic field from 3.6 T to 4.1 T, the local pairing strength Δ remains stable and the coefficient C decreases. It is reasonable that the influence of DOS term (this term takes into account the decrease of quasiparticle DOS due to formation of local fluctuating Cooper pairs) becomes smaller in higher magnetic field since the fluctuating Cooper pairs are reduced. The normal state resistance is plotted in Fig. R1, which exhibits the behavior of weakly localized metallic state. We have added the above explanation in the revised Supplementary Note 1.

Fig. R1 (a) The experimental $R_s(T)$ curves and the theoretical fitting curves of 4-ML Pb film from 3.6 T to 4.1 T. The theoretical curves are quantitatively consistent with the experimental data of 4-ML Pb film. (b) Theoretical normal state resistance $R_n(T)$ curves at various magnetic fields.

Table R1 The fitting parameters at different magnetic fields from 3.6 T to 4.1 T

B (T)	α	β	C (Ω^{-1})	Δ/k_B (K)	σ_0 (Ω^{-1})	d
3.6	0.518	-0.233	9.71×10^{-5}	10.2	5.11×10^{-4}	0.519
3.7	0.518	-0.218	9.47×10^{-5}	10.3	5.00×10^{-4}	0.529
3.8	0.525	-0.205	9.33×10^{-5}	10.2	4.51×10^{-4}	0.583
3.9	0.527	-0.194	9.09×10^{-5}	10.2	3.80×10^{-4}	0.660
4.0	0.528	-0.195	8.72×10^{-5}	10.3	2.92×10^{-4}	0.755
4.1	0.515	-0.171	8.27×10^{-5}	10.3	2.57×10^{-4}	0.789

Comment: (6) After clarifying the above questions, there remains a problem. In the case that $R(T)$ can be reproduced by the thermal fluctuation theory, B_c with the minima in $R(T)$ is not the boundary between the superconducting state and weakly localized metal state, but is simply determined by the balance between the two kinds of terms in eq. (4), which decreases and increases with decreasing temperature, respectively. On the other hand, the authors performed finite size scaling analyses based on the quantum fluctuation model by using B_c . The fact that the electrical

resistance in the same temperature and magnetic field regions is analyzed by different kinds of approaches, thermal and quantum fluctuation, may cause contradiction.

Response: We thank reviewer #2 for pointing out this important question. Experimentally, it is a common way to determine the phase boundary by the minima and maximum of the $R_s(T)$ curves (*Nature Materials* **12**, 542(2013)). In 2D systems with pronounced superconducting fluctuation, the phase boundary between superconducting state and weakly localized metal state deviates from the mean field theory. The anomalous phase boundary in the low temperature regime mainly depends on the AL-type and MT-type superconducting fluctuation, while the temperature dependence of the thermally activated behavior becomes negligible at low temperatures.

Moreover, the superconducting fluctuation contains both thermal and quantum fluctuations. With decreasing temperature, the thermal fluctuation is gradually suppressed and the quantum fluctuation plays a key role in determining the sheet resistance and the phase boundary. For the anomalous quantum Griffiths singularity, disorder induced quantum fluctuation also becomes very important in the low temperature regime and changes the characteristic of the critical exponent $z\nu$. Therefore, at low temperatures, the scaling analysis and the theoretical fitting of phase boundary mainly depends on the quantum fluctuation. We have added related discussion to clarify this question in the revised Supplementary Note 1.

***Comment:** (7) The reentrant behavior of the resistance has been already reported in the amorphous films (ref. 6) as well as TiN films (ref 39). In addition to the low mobility as the authors indicate, the similarities and the differences between the precious reports and the present data should be discussed. At least, in ref.6, the homogeneity of the sample is stressed, which seems to be a common feature.*

Response: We thank Reviewer #2 for his/her constructive suggestion. Although similar reentrant behaviors have been reported in superconductor insulator/metal transition (SIT/SMT) of superconducting thin films, we would like to point out two differences that distinguish our work from these earlier works. In the ultrathin Pb film,

strong SOI can influence the parameter α and β of the superconducting fluctuation and hence change the extent of the superconducting fluctuations, which is different from previous studies on the reentrant behavior (*Phys. Rev. Lett.* **65**, 927-930 (1990), *Phys. Rev. B* **69**, 024505 (2004)). Furthermore, as shown in Ref. 6 (*Phys. Rev. Lett.* **65**, 927-930 (1990)), the critical exponent $z\nu$ for the amorphous InO_x film is a constant around 1.3, indicating that the quantum Griffiths singularity is not observed. The homogenous disorder in amorphous InO_x film gives rise to SIT with critical resistance near $\frac{h}{4e^2}$, while in our case the disorder gives rise to non-homogenous rare region near the SMT with critical resistance much smaller than $\frac{h}{4e^2}$ and the rare regions are responsible for the quantum Griffiths singularity (*Phys. Rev. B* **79**, 024401 (2009) and *Science* **350**, 542 (2015)). We have added the discussion in the revised Supplementary Note 1.

Comment: (8) *In this manuscript, the authors only show the magnetoresistance near the onset. Many readers would like to know the whole picture of resistive transition in magnetic fields.*

Response: We thank Reviewer #2 for the helpful suggestion. The magnetoresistance in large magnetic field regime has been added to the revised supplementary Fig. 2 and Fig. R4.

Fig. R4. (a) The magnetoresistance of 4 ML Pb film at different temperatures from 0.5 K to 7.0 K. (b) Close-up of the same data near the crossing region of $R_s(B)$ curves.

Reviewer #3 (Remarks to the Author):

Comment: *The manuscript "Anomalous quantum Griffiths singularity in ultrathin crystalline lead films" contains interesting experimental data. These data are interpreted using a theoretical concept of quantum Griffiths singularity. The problem is that this phenomenon probably does not exist in disordered superconducting films. One can find a discussion of the situation in the recent review <https://arxiv.org/abs/1712.07215>*

On the other hand, it is possible that in an intermediate temperature range experimental data can mimic quantum Griffiths singularity. In my opinion this question should be clarified before publishing of the article.

Response: We gratefully thank Reviewer #3 for the careful reading of our manuscript and for his/her positive comment "The manuscript ... contains interesting experimental data".

In the review paper (arXiv 1712.07215), the authors suggest true quantum Griffiths state may not exist due to the reason that "Gapless degrees of freedom associated with the surrounding metallic state can only penetrate at most a distance ξ_0 into the cluster. Therefore, in the end, the coupling to the heat-bath can at most grow in proportion to the perimeter (surface area in 3D) of the cluster". However, based on the Bardeen-Cooper-Schreiffer (BCS) theory, the superconducting coherence length $\xi_0 = \frac{\hbar v_F}{\pi \Delta(0)}$ (where $\Delta(0)$ is the superconducting gap and v_F is Fermi velocity, Tinkham, M. *Introduction to superconductivity*. Courier Corporation (2004)) diverges with the superconducting gap $\Delta(0)$ tending to zero near the phase boundary. (Cooper, L. N. *BCS: 50 years*, World scientific (2011), Chap. 11). Therefore, it is possible to meet the requirement of quantum Griffiths singularity that ξ_0 should be larger than the scale of the rare regions. Moreover, based on comments of Reviewer #2, we are aware of a highly relevant paper on the two-dimensional superconducting fluctuation (*Phys. Rev. B* **84**, 104510 (2011)). In this work, the authors analyze the

coherence cluster of fluctuating Cooper pairs with size $\xi_{QF} \sim \xi_{BCS}(H=0)/\sqrt{\frac{H_{c2}(0)}{H-H_{c2}(0)}}$, which diverges near the quantum critical point $B_{c2}(0)$ (Fig.4 and related discussion in *Phys. Rev. B* **84**, 104510 (2011)) and thus is consistent with the prerequisite of quantum Griffiths singularity.

On the other hand, previous experiments on various 2D superconducting systems report $z\nu$ increases rapidly and tends to diverge with decreasing temperature in the ultralow temperature regime (*Science* **350**, 542-545 (2015), *Nature Communications* **9**, 778 (2018)). Indeed, a truly divergent $z\nu$ has not been experimentally demonstrated due to the requirement of extremely low temperature and a very high measurement resolution of resistance. However, the experimental value of $z\nu$ in these systems including the ultrathin Pb films changes in a large scale and does not have a trend to saturate even at the lowest temperatures we can achieve in dilution refrigerator, which can only be explained by the theory of quantum Griffiths singularity. We have added the related discussion in the revised Supplementary Note 2.

Reviewers' comments:

Reviewer #1 (Remarks to the Author):

I am satisfied with the revised version of the manuscript, and now recommend for publication.

Reviewer #2 (Remarks to the Author):

The authors have addressed most of my concerns and made major revision in the discussion part (main text) and in Supplementary Information. However there still remains the question about the relation between theoretical fitting by fluctuation theory and the Griffiths scaling behavior. As I point out in my original comment (6), the authors reproduce $R(T)$ by using the fluctuation theories (Ref. 17 and 20), which do not include the effect of disorder causing the Griffiths singularity. If the authors claim that they succeed to reproduce experimental $R(T)$ completely with the fluctuation theory, it means that measured $R(T)$ can be explained in the scheme without the effect of disorder and the concept of the Griffiths singularity is not necessary. Nevertheless, the behavior of the critical exponent derived from the scaling law indicate the characteristic of the Griffiths singularity, which is most interesting point of this paper. The authors should resolve this contradiction. If the both the models of quantum/thermal fluctuation and the quantum Griffiths singularity (effect of the disorder) is reflected to the data, it is natural to consider that the onset of reentrant behavior of $R(T)$ is explained by the fluctuation theory, but at the low temperature region, where the effect of disorder is significant, $R(T)$ deviates from it due to the effect of disorder. It seems that the unusual theoretical normal state resistance in Supplementary Figure 1(b), which increase with field (usually decrease with field in weak localization system) and show much lower values than those of the zero-field normal resistance (red dashed line in Fig. 1(a)), might be related to this question.

Reviewer #3 (Remarks to the Author):

As I have mentioned in my original report, the article contains interesting experimental data. On the other hand, I disagree with both the original theoretical interpretation of the data and with author's response to my comment. Since I do not see a possibility to reach an agreement on the theoretical part of the paper I leave the decision to the editor.

a) I do not see how the article of A. Glatz mentioned in the response is relevant to the issue of the existence of the Griffiths phase. In this article small superconducting fluctuations corrections to the conductivity were calculated using conventional perturbation theory which ignores the possibility of rare events. So it ignores the possibility of the Griffiths phase.

It is a separate question whether the results of this article can be used for interpreting the magnetoresistance and the temperature dependence of the the resistance discussed in the present manuscript.

b) The arguments presented by the authors in their response actually indicate that they observe an intermediate asymptotic rather than the Griffiths phase. The existence of the Griffiths phase is based on the assumption that the susceptibility of the superconducting grains is exponential in the volume of the grains in the limit of the infinite volume. As the authors correctly indicated in their response, it may be exponential in the volume as long as the radius of the grains is smaller than the coherence length. This however is not the case at larger radii, indicating the absence of the genuine Griffiths phase in a superconductor-metal transition.

In the following, the referee's original comments are shown by blue italic characters.

Our responses are shown by black characters.

Reviewer #1 (Remarks to the Author):

I am satisfied with the revised version of the manuscript, and now recommend for publication.

Response: We thank Reviewer #1 for his/her approval of our reply and modifications.

We also appreciate his/her recommendation for publication.

Reviewer #2 (Remarks to the Author):

The authors have addressed most of my concerns and made major revision in the discussion part (main text) and in Supplementary Information. However there still remains the question about the relation between theoretical fitting by fluctuation theory and the Griffiths scaling behavior. As I point out in my original comment (6), the authors reproduce $R(T)$ by using the fluctuation theories (Ref.17 and 20), which do not include the effect of disorder causing the Griffiths singularity. If the authors claim that they succeed to reproduce experimental $R(T)$ completely with the fluctuation theory, it means that measured $R(T)$ can be explained in the scheme without the effect of disorder and the concept of the Griffiths singularity is not necessary. Nevertheless, the behavior of the critical exponent derived from the scaling law indicate the characteristic of the Griffiths singularity, which is most interesting point of this paper. The authors should resolve this contradiction. If the both the models of quantum/thermal fluctuation and the quantum Griffiths singularity (effect of the disorder) is reflected to the data, it is natural to consider that the onset of reentrant behavior of $R(T)$ is explained by the fluctuation theory, but at the low temperature region, where the effect of disorder is significant, $R(T)$ deviates from it due to the effect of disorder. It seems that the unusual theoretical normal state resistance in Supplementary Figure 1(b), which increase with field (usually decrease with field in weak localization system) and show much lower values than those of the zero-field normal resistance (red dashed line in Fig. 1(a)), might be related to this question.

Response: We thank Reviewer #2 for his/her approval of our reply and modifications. We fully agree with Reviewer #2's insightful comment that the divergence of $z\nu$ as a feature of quantum Griffiths singularity cannot be reproduced by only considering the superconducting fluctuation theory (*JETP Lett.* 77,424(2003)). We also appreciate his/her constructive comments and helpful suggestions during the reviewing process. As shown in Fig. R1(a), the experimental $R_s(T)$ curves from 3.6 T to 4.1 T are consistent with the theoretical fitting based on the superconducting fluctuation theory,

indicating the experimental phase boundary of superconductor-metal transition at the fields above 3.6 T (corresponding to the temperatures above 1 K) can be well explained in the frame of superconducting fluctuation effect (Fig. R1(b)).

Nevertheless, when approaching lower temperature and lower magnetic field regime along the phase boundary, the disorder induced higher order corrections beyond the fluctuation effect become pronounced, which finally gives rise to the anomalous quantum Griffiths singularity. It is noteworthy to mention that the experimental $R_s(T)$ curve at 3.5 T begins to deviate from the theoretical formula of fluctuation effect, especially at low temperatures below 1 K, which represents the higher order corrections induced by disorder effect when approaching the critical point. Thus, both the fluctuation effect and the disorder induced quantum Griffiths singularity are important to understand the SMT in our system. The above discussion has been added to the revised manuscript and Supplementary Information.

Moreover, the normal state resistance for the theoretical fitting in Fig. R1(d) increases with increasing magnetic field, which can be attributed to the influence of spin-orbit coupling similar to systems with weak anti-localization. The temperature dependence of the normal state resistance may relate to the electron-electron interaction. The reason for the lower normal resistance compared to the experimental total resistance may be two-fold. First, this feature may originate from the higher order corrections not included in the fluctuation effect, as pointed out by Reviewer #2. Second, this feature is also related to the fluctuation effect in the quantum fluctuation regime. As shown in region IV of Table I in *Phys. Rev. B* **84**, 104510 (2011), the total correction to conductivity from fluctuation is negative.

Fig. R1 (a) The experimental $R_s(T)$ curves and the theoretical fitting curves of 4-ML Pb film from 3.6 T to 4.1 T. (b) The theoretical phase boundary from 3.6 T to 4.1T, which is consistent with the experimental data. (c) The theoretical fitting curve deviates from the experimental $R_s(T)$ curve of 3.5 T, especially in the ultralow temperature regime, which may result from the influence of disorder. (d) The normal state resistance $R_n(T)$ curves at various magnetic fields for theoretical fitting.

Reviewer #3 (Remarks to the Author):

As I have mentioned in my original report, the article contains interesting experimental data. On the other hand, I disagree with both the original theoretical interpretation of the data and with author's response to my comment. Since I do not see a possibility to reach an agreement on the theoretical part of the paper I leave the decision to the editor.

Response: We thank Reviewer #3 for the high evaluation of our experimental result by emphasizing that “As I have mentioned in my original report, the article contains interesting experimental data.” We also appreciate his/her helpful comments during the reviewing process.

a) I do not see how the article of A. Glatz mentioned in the response is relevant to the issue of the existence of the Griffiths phase. In this article small superconducting fluctuations corrections to the conductivity were calculated using conventional perturbation theory which ignores the possibility of rare events. So it ignores the possibility of the Griffiths phase.

It is a separate question whether the results of this article can be used for interpreting the magnetoresistance and the temperature dependence of the the resistance discussed in the present manuscript.

Response: We thank Reviewer #3 for raising this issue. We agree with Reviewer #3 that the fluctuation effect cannot lead to the quantum Griffiths phase. Our theoretical model based on the superconducting fluctuation (*Phys. Rev. B* 63, 174506 (2001), *Phys. Rev. B* 84, 104510 (2011)) is used to explain the buckle shape of the phase boundary in Fig. R1(b). In fact, it works well at relatively high temperature region and gives a good fitting to the experimental data above 3.6 T (corresponding to temperatures above 1 K). Nevertheless, when approaching the lower temperature and lower field regime along the phase boundary, the disorder induced higher order corrections beyond the fluctuation effect becomes pronounced, which finally gives rise to the quantum Griffiths phase. Subsequently, the experimental $R_s(T)$ curve at

3.5 T begins to deviate from the theoretical formula, especially at low temperatures below 1 K, which may result from the disorder effect and the Griffiths phase (Fig. R1(c)). Therefore, the superconducting fluctuation theory gives a satisfactory understanding to the reentrant behavior and the buckle shape of phase boundary above 1 K in the 4-ML Pb film, while the disorder induced higher order corrections beyond the fluctuation effect become dominant and gives rise to the quantum Griffiths phase at lower temperatures. We have added the above discussion in the revised manuscript and Supplementary Information.

Fig. R1 (a) The experimental $R_s(T)$ curves and the theoretical fitting curves of 4-ML Pb film from 3.6 T to 4.1 T. (b) The theoretical phase boundary from 3.6 T to 4.1T, which is consistent with the experimental data. (c) The theoretical fitting curve deviates from the experimental $R_s(T)$ curve of 3.5 T, especially in the ultralow temperature regime, which may result from the influence of disorder. (d) The normal state resistance $R_n(T)$ curves at various magnetic fields for theoretical fitting.

b) The arguments presented by the authors in their response actually indicate that they observe an intermediate asymptotic rather than the Griffiths phase. The existence of the Griffiths phase is based on the assumption that the susceptibility of the superconducting grains is exponential in the volume of the grains in the limit of the infinite volume. As the authors correctly indicated in their response, it may be exponential in the volume as long as the radius of the grains is smaller than the coherence length. This however is not the case at larger radii, indicating the absence of the genuine Griffiths phase in a superconductor-metal transition.

Response: We greatly appreciate the comments of Reviewer #3 on this in-depth theoretical issue. As we mentioned in the previous response, the coherence length can be divergent near the phase boundary (Tinkham, M. *Introduction to superconductivity*. Courier Corporation (2004), Cooper, L. N. *BCS: 50 years*, World scientific (2011), Chap. 11). This statement is valid along the entire phase boundary even when approaching zero temperature (the radius of the rare region keeps increasing and finally approaches infinite volume at this process). Therefore, the coherence length is possible to be larger than the scale of the rare regions (even for a large radius at ultralow temperatures) when approaching infinite randomness quantum critical point and gives rise to the quantum Griffiths singularity. In the experimental aspect, we would like to emphasize that the observed value of $z\nu$ in the 4-ML Pb films changes in a large scale and $z\nu$ does not have a trend of saturation even at the lowest temperatures achievable in our dilution refrigerator. To our knowledge, these observations can only be explained in the framework of quantum Griffiths singularity. We have added the above discussion in the revised Supplementary Information.